# Efficient multi-prompt evaluation of LLMs

**Felipe Maia Polo**[1][*], **Ronald Xu**[2,6], **Lucas Weber**[3], **Mírian Silva**[4,5,6], **Onkar Bhardwaj**[5,6]
**Leshem Choshen**[2,5,6], **Allysson Flavio Melo de Oliveira**[5,6], **Yuekai Sun**[1], **Mikhail Yurochkin**[5,6]
[1]University of Michigan, [2]MIT, [3]University Pompeu Fabra, [4]Federal University of Minas Gerais
[5]IBM Research, [6]MIT-IBM Watson AI Lab

## Abstract

Most popular benchmarks for comparing LLMs rely on a limited set of prompt templates, which may not fully capture the LLMs' abilities and can affect the reproducibility of results on leaderboards. Many recent works empirically verify prompt sensitivity and advocate for changes in LLM evaluation. In this paper, we consider the problem of estimating the performance *distribution* across many prompt variants instead of finding a single prompt to evaluate with. We introduce PromptEval, a method for estimating performance across a large set of prompts borrowing strength across prompts and examples to produce accurate estimates under practical evaluation budgets. The resulting distribution can be used to obtain performance quantiles to construct various robust performance metrics (e.g., top 95% quantile or median). We prove that PromptEval consistently estimates the performance distribution and demonstrate its efficacy empirically on three prominent LLM benchmarks: MMLU, BIG-bench Hard, and LMentry; for example, PromptEval can accurately estimate performance quantiles across 100 prompt templates on MMLU with a budget equivalent to two single-prompt evaluations. Moreover, we show how PromptEval can be useful in LLM-as-a-judge and best prompt identification applications.[2]

## 1 Introduction

In recent years, the rapid progress of large language models (LLMs) has significantly influenced various fields by enhancing automated text generation and comprehension. As these models advance in complexity and functionality, a key challenge that arises is their robust evaluation [Perlitz et al., 2023]. Common evaluation methods, which often rely on a single or limited number of prompt templates, may not adequately reflect the typical model's capabilities [Weber et al., 2023b]. Furthermore, this approach can lead to unreliable and inconsistent rankings on LLM leaderboards, as different models may perform better or worse depending on the specific prompt

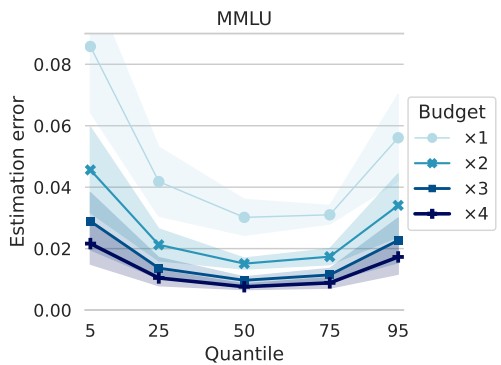

Figure 1: Average estimation error for performance quantiles across 100 templates given a limited budget (in multiples of one-template MMLU evaluations).

---

[*]Corresponding author. E-mail: felipemaiapolo@gmail.com

[2]Our code can be found in https://github.com/felipemaiapolo/prompteval and the MMLU data can be found in https://huggingface.co/PromptEval. PromptEval is also integrated into PromptBench [Zhu et al., 2024]; please check https://github.com/microsoft/promptbench.

38th Conference on Neural Information Processing Systems (NeurIPS 2024).

template used. An ideal evaluation framework should minimize dependence on any single prompt template and instead provide a holistic summary of performance across a broad set of templates. Mizrahi et al. [2023], for example, suggests using summary statistics, such as the average performance across many templates, as a way to compare the abilities of different LLMs. However, the main drawback of this method is the high computational cost when dealing with numerous templates and examples.

We introduce PromptEval, a method for efficient multi-prompt evaluation of LLMs. With a small number of evaluations, PromptEval estimates performance across a large and *given* pool of different prompt templates. Our approach is grounded in robust theoretical foundations and utilizes well-established models from the fields of educational assessment and psychometrics, such as Item Response Theory (IRT) [Cai et al., 2016, Van der Linden, 2018, Brzezińska, 2020, Lord et al., 1968]. Our method is based on an IRT model that allows borrowing strength across examples and prompt templates to produce accurate estimates of all considered prompts with an evaluation budget comparable to evaluating a single prompt. In Figure 1, we demonstrate the ability of our method to jointly estimate various performance quantiles across 100 prompt templates with an evaluation budget ranging from one to four times of a conventional single-prompt evaluation on MMLU [Hendrycks et al., 2020].

Performance distribution across prompts can be used to accommodate various contexts when comparing LLMs [Choshen et al., 2024]. For example, it can be used to compute the mean performance as suggested by Mizrahi et al. [2023]. One can also use performance distributions directly to compare LLMs via various notions of stochastic dominance for risk-sensitive scenarios [Nitsure et al., 2023]. Here we primarily focus on the full distribution and its quantiles as they provide a flexible statistic that can inform decisions in varying contexts. For instance, a typical model performance corresponds to a median (50% quantile), 95% quantile can be interpreted as performance achievable by an expert prompt engineer, while 5% quantile is of interest in consumer-facing applications to quantify low-end performance for a user not familiar with prompt engineering. We also demonstrate (§6) how our method can be used to account for prompt sensitivity in the LLM-as-a-judge framework [Li et al., 2023] and to do best prompt identification [Shi et al., 2024].

Our main contributions are:

- We propose (§3) a novel method called PromptEval which permits efficient multi-prompt evaluation of LLMs for a *given* pool of prompt templates with a limited number of evaluations. Moreover, we theoretically show (§4) that PromptEval has desirable statistical properties such as consistency in estimating performance distribution and its quantiles.
- We practically demonstrate (§5) efficacy of PromptEval in estimating performance across 100+ prompts and finding the best-performing prompt for various LLMs using data derived from three popular benchmarks: MMLU [Hendrycks et al., 2020], BIG-bench Hard (BBH) [Suzgun et al., 2022], and LMentry [Efrat et al., 2022].
- We show (§6) how PromptEval can be applied to account for prompt sensitivity in the LLM-as-a-judge framework and to identify the best prompt in a large pool of options.
- We conduct the first large-scale study of prompt sensitivity of 15 popular open-source LLMs on MMLU. We present our findings based on evaluating 100 prompt templates in Section 7 and release the evaluation data.

## 1.1 Related work

**LLMs' sensitivity to prompt templates** The sensitivity of Large Language Models (LLMs) to the prompts is well-documented. For example, Sclar et al. [2023] revealed that subtle variations in prompt templates in few-shot settings can lead to significant performance discrepancies among several open-source LLMs, with differences as large as 76 accuracy points in tasks from the SuperNaturalInstructions dataset [Wang et al., 2022]. Additionally, they report that the performance of different prompt templates tends to correlate weakly between models. This finding challenges the reliability of evaluation methods that depend on a single prompt template. To measure LLMs sensitivity, the researchers suggested calculating a "performance spread," which represents the difference between the best and worst performances observed. Mizrahi et al. [2023] conducted a complementary analysis using state-of-the-art models and subsets of BigBench and LMentry [Srivastava et al., 2022, Efrat et al., 2022]. The authors arrive at similar conclusions with respect to LLMs' sensitivity to the used

prompt templates and empirically showed that the LLM ranking considering different formats are usually weakly or intermediately correlated with each other. As a solution to the lack of robustness in LLM evaluation, the authors propose the use of summary statistics, as the average performance, for LLM evaluation. Some other works, *e.g.*, Voronov et al. [2024], Weber et al. [2023b,a], show that even when in-context examples are given to the models, the prompt templates can have a big impact on the final numbers, sometimes reducing the performance of the strongest model in their analyses to a random guess level [Voronov et al., 2024]. In a different direction, Shi et al. [2024] acknowledges that different prompt templates have different performances and proposes using best-arm-identification to efficiently select the best template for an application at hand. One major bottleneck is still on how to efficiently compute the performance distribution for LLMs over many prompt templates; we tackle this problem.

**Efficient evaluation of LLMs** The escalating size of models and datasets has led to increased evaluation costs. To streamline evaluations, Ye et al. [2023b] considered minimizing the number of *tasks* within Big-bench [Srivastava et al., 2022]. Additionally, Perlitz et al. [2023] observed that evaluations on HELM [Liang et al., 2022] rely on diversity across datasets, though the quantity of examples currently utilized is unnecessarily large. Perlitz et al. [2023] also highlighted the problems in evaluating with insufficient prompts and called to evaluate on more, suggesting evaluating the typical behavior by sampling prompts and examples together by employing stratified sampling, where subscenarios give the strata; in our work, we also apply stratification but consider prompt templates and examples to give the strata. To accelerate evaluations for classification tasks, Vivek et al. [2023] suggested clustering evaluation examples based on model confidence in the correct class. More recently, Maia Polo et al. [2024] empirically showed that it is possible to shrink the size of modern LLM benchmarks and still retain good estimates for LLMs' performances. Similarly (and in parallel to this work) Ashury-Tahan et al. [2024] recognized unlabeled examples that better distinguish between models or prompts, by analyzing model outputs on them, hence saving costly annotation for them. Despite these advancements in streamlining LLM evaluations, there are no other works that propose a general and efficient method to estimate the benchmark performance of LLMs across prompt templates to the best of our knowledge.

**Item response theory (IRT)** IRT [Cai et al., 2016, Van der Linden, 2018, Brzezińska, 2020, Lord et al., 1968] is a collection of statistical models initially developed in psychometrics to assess individuals' latent abilities through standardized tests but with increasing importance in the fields of artificial intelligence and natural language processing (NLP). For example, Lalor et al. [2016] used IRT's latent variables to measure language model abilities, Vania et al. [2021] applied IRT to benchmark language models and examine the saturation of benchmarks, and Rodriguez et al. [2021] explored various uses of IRT with language models, including predicting responses to unseen items, categorizing items by difficulty, and ranking models. Recently, Maia Polo et al. [2024], Shabtay et al. [2024] suggested using IRT for efficient LLM performance evaluation; both works used the Performance-IRT (pIRT) estimator to evaluate LLMs. PromptEval is built upon pIRT.

## 2  Problem statement

In this section, we describe the setup we work on and what our objectives are. Consider that we want to evaluate a large language model (LLM) in a certain dataset composed of $J$ examples (also known as questions or items in the literature) and each one of the examples is responded to by the LLM through prompting; we assume that there exists $I$ different prompt templates that can be used to evaluate the LLM. After the prompt template $i \in \mathcal{I} \triangleq [I]$ and example $j \in \mathcal{J} \triangleq [J]$ are channelled through the LLM, some grading system generates a correctness score $Y_{ij} \in \{0, 1\}$, which assumes 1 when the prompt template $i$ and example $j$ jointly yield a correct response and 0 otherwise[3]. For each one of the prompt templates $i \in \mathcal{I}$, we can define its performance score as

$$S_i \triangleq \tfrac{1}{J} \sum_{j \in \mathcal{J}} Y_{ij}.$$

The performance scores $S_i$'s can have a big variability, making the LLM evaluation reliant on the prompt choice. To have a comprehensive evaluation of the LLM, we propose computing the full *distribution of performances* and its corresponding quantile function, *i.e.*,

$$F(x) \triangleq \tfrac{1}{I} \sum_{i \in \mathcal{I}} \mathbb{1}_{[S_i, \infty)}(x) \quad \text{and} \quad Q(p) \triangleq \inf\{x \in \mathbb{R} : F(x) \geq p\}. \tag{2.1}$$

---

[3]In some cases, the correctness score may be a bounded number instead of binary – see Appendix B.

The main challenge in obtaining this distribution is that it can be very expensive since the exact values for the performance scores $S_i$'s require $I \cdot J$ evaluations. In this paper, we assume that only a small fraction of evaluations is available, *e.g.*, $< 5\%$ of the total number of possible $I \cdot J$ evaluations, but we still aim to accurately estimate the performance distribution and its quantiles. More concretely, we assume the correctness scores $Y_{ij}$'s are only evaluated for a small set of indices $\mathcal{E} \subseteq \mathcal{I} \times \mathcal{J}$; in compact notation, we define $Y_\mathcal{E} \triangleq \{Y_{ij}\}_{(i,j) \in \mathcal{E}}$. Here, the letter $\mathcal{E}$ stands for *evaluations*. Using the observed data $Y_\mathcal{E}$, our main objective is to estimate the performance scores distribution $F$ (resp. quantile function $Q$), *i.e.*, computing a function $\hat{F}$ (resp. $\hat{Q}$) that is *close* to $F$ (resp. $Q$).

## 3 Performance distribution and quantiles estimation

We propose borrowing strength across prompt templates and examples to produce accurate estimates for the performance distribution and its quantile function. To achieve that, we need a model for the correctness scores $Y_{ij}$'s that allows leveraging patterns in the observed data and estimators for individual $S_i$'s. We start this section by first introducing a general model for $Y_{ij}$'s and then we introduce our estimators for the performance distribution and quantile functions.

### 3.1 The correctness model

We assume the observations $Y_{ij}$'s are independently sampled from a Bernoulli model parameterized by prompt/example-specific parameters. That is, we assume

$$Y_{ij} \sim \text{Bernoulli}(\mu_{ij}), \tag{3.1}$$

where $\mu_{ij}$ denotes the mean of the Bernoulli distribution specific to prompt format $i$ and example $j$. We can write $\mu_{ij} = \mu(\theta_i, \beta_j)$, where $\theta_i$'s are prompt-specific parameters, $\beta_j$'s are example-specific parameters and $\mu$ is a function that maps those parameters to the Bernoulli mean. This probabilistic model is very general and comprehends factor models such as the large class of Item Response Theory (IRT) models [Cai et al., 2016, Van der Linden, 2018, Brzezińska, 2020, Lord et al., 1968]; as we will see, our model can be seen as a general version of an IRT model. For generality purposes, we assume that the parameters $\theta_i$'s and $\beta_j$'s can be written as functions of prompt-specific ($x_i$'s) and example-specific ($z_j$'s) vectors of covariates. That is, we assume $\theta_i = f_\psi(x_i)$ or $\beta_j = g_\gamma(z_j)$, where $\psi$ and $\gamma$ are global parameters that can be estimated. These covariates can be, for example, embeddings of prompt templates in the case of $x_i$'s and some categorization or content of each of the examples in the case of $z_j$'s. In this work, we adopt $\mu(\theta_i, \beta_j) = \sigma(\theta_i - \beta_j) = \sigma(f_\psi(x_i) - g_\gamma(z_j))$, where $\sigma$ denotes the standard logistic function and the functions $f_\psi$ and $g_\gamma$ have their image in $\mathbb{R}$. That is, our model assumes that

$$\mathbb{P}(Y_{ij} = 1; \psi, \gamma) = \sigma\big(f_\psi(x_i) - g_\gamma(z_j)\big) \triangleq \frac{1}{1 + \exp[-(f_\psi(x_i) - g_\gamma(z_j))]}. \tag{3.2}$$

The functions $f_\psi$ and $g_\gamma$ can be represented with neural networks. On the simpler side, one could just assume $f_\psi$ and $g_\gamma$ are linear, that is, $\theta_i = \psi^\top x_i$ or $\beta_j = \gamma^\top z_j$; this formulation is known as the linear logistic test model in psychometrics [Fischer, 1973, De Boeck, 2004]. We consider that, in some cases, a constant can be embedded in $x_i$ in order to include an intercept in the model. When $x_i$ and $z_j$ are one-hot encoded vectors, *i.e.*, vector of zeros but with 1's on the entries $i$ and $j$, the model in (3.2) reverts to a popular IRT model known as the Rasch model [Georg, 1960, Chen et al., 2023], which is widely used in fields such as recommendation systems [Starke et al., 2017] and educational testing [Clements et al., 2008]. One major limitation of the basic Rasch model is that the number of parameters is large, compromising the quality of the estimates for $\psi$ and $\gamma$ when either the number of prompt formats $I$ or the number of examples $J$ is large and $|\mathcal{E}|$ is small, *i.e.*, only a few evaluations are carried out. This degradation in the quality of the estimates can directly affect the quality of the performance distribution estimates. Finally, we fit the parameters $\psi$ and $\gamma$, obtaining the estimates $\hat{\psi}$ and $\hat{\gamma}$, by maximizing the log-likelihood of the observed data (negative cross-entropy loss), *i.e.*,

$$(\hat{\psi}, \hat{\gamma}) \in \arg\max_{\psi, \gamma} \sum_{(i,j) \in \mathcal{E}} Y_{ij} \log \mathbb{P}(Y_{ij} = 1; \psi, \gamma) + (1 - Y_{ij}) \log\big(1 - \mathbb{P}(Y_{ij} = 1; \psi, \gamma)\big). \tag{3.3}$$

Realize that fitting the model with linear/affine $f_\psi$ and $g_\gamma$, including the Rasch model case[4], reduces to fitting a logistic regression model with $x_i$ and $z_j$ as the covariates. This observation highlights that

---

[4]For a detailed fitting procedure in the Rasch model case, please check Chen et al. [2023].

| **Algorithm 1:** PromptEval | **Algorithm 2:** Two-way balanced sampling |
|---|---|
| **1 Input:** (i) $Y_{\mathcal{E}}$, (ii) covariates $x_i$'s and $z_j$'s. | **1 Input:** (i) sets $\mathcal{I}$ and $\mathcal{J}$, (ii) budget $B$. |
| **2 Output:** Estimates for the performances distribution and its quantile function (2.1). | **2 Output:** Observed indices $\mathcal{E}$. |
| | **3** Initialize $\mathcal{E} = \{\}$. |
| **3** Fit $\psi$ and $\gamma$ using observed scores $Y_{\mathcal{E}}$ and covariates $x_i$'s and $z_j$'s (3.3). | **4 for** $b = 0$ *to* $B - 1$ **do** |
| **4** For each $i \in \mathcal{I}$, compute $\hat{S}_i = \hat{\mathbb{E}}[S_i \mid Y_{\mathcal{E}}]$ (3.4). | **5**      Among $i \in \mathcal{I}$ with the least number of evaluations, randomly pick one of them and call it $\hat{i}$. |
| **5** Compute estimates $$\hat{F}(\cdot) \triangleq \frac{1}{I} \sum_{i \in \mathcal{I}} \mathbb{1}_{[\hat{S}_i, \infty)}(\cdot)$$ $$\hat{Q}(\cdot) \triangleq \inf\{x \in \mathbb{R} : \hat{F}(x) \geq \cdot\}$$ **return** $\hat{F}$ and $\hat{Q}$. | **6**      Among $j \in \mathcal{J}$ such that $(\hat{i}, j) \notin \mathcal{E}$, randomly pick $\hat{j}$ from the ones with the least number of evaluations. |
| | **7**      Update $\mathcal{E} \leftarrow \mathcal{E} \cup \{(\hat{i}, \hat{j})\}$ |
| | **8 return** $\mathcal{E}$. |

the fitting process is expected to be very cheap in practice. For example, in our experiments, we fit logistic regression models in datasets with less than 2k samples and a couple of hundred (or a few thousand) columns, which is performed in a few seconds by a modern laptop. We include some more comments on the computational complexity of our method in Appendix C.

### 3.2 Performance distribution and quantiles estimation using the correctness model

The model in (3.1) can be naturally used for performance estimation. That is, after observing $Y_{\mathcal{E}}$, the best approximation (in the mean-squared-error sense) for the performance of prompt format $i \in \mathcal{I}$, $S_i$, is given by the following conditional expectation

$$\mathbb{E}[S_i \mid Y_{\mathcal{E}}] = \frac{\lambda_i}{|\mathcal{J}_i|} \sum_{j \in \mathcal{J}_i} Y_{ij} + \frac{1 - \lambda_i}{|\mathcal{J} \setminus \mathcal{J}_i|} \sum_{j \notin \mathcal{J}_i} \mu_{ij}$$

where $\mathcal{J}_i \triangleq \{j \in \mathcal{J} : (i, j) \in \mathcal{E}\}$ and $\lambda_i = |\mathcal{J}_i|/J$. In practice, computing $\mathbb{E}[S_i \mid Y_{\mathcal{E}}]$ is impossible because the parameters $\theta_i$'s and $\beta_j$'s are unknown. We can, however, use a plug-in estimator for the conditional expectation using their maximum likelihood estimators, changing $\mu_{ij}$ for $\sigma(f_{\hat{\psi}}(x_i) - g_{\hat{\gamma}}(z_j))$:

$$\hat{\mathbb{E}}[S_i \mid Y_{\mathcal{E}}] = \frac{\lambda_i}{|\mathcal{J}_i|} \sum_{j \in \mathcal{J}_i} Y_{ij} + \frac{1 - \lambda_i}{|\mathcal{J} \setminus \mathcal{J}_i|} \sum_{j \notin \mathcal{J}_i} \sigma(f_{\hat{\psi}}(x_i) - g_{\hat{\gamma}}(z_j)). \tag{3.4}$$

The basic version of this estimator, when no elaborate covariates (*e.g.*, embeddings) are included, is known as the Performance-IRT (pIRT) estimator [Maia Polo et al., 2024]. We can apply our extended version of pIRT, which we call X-pIRT, to estimate the performance distribution across prompt templates. After observing $Y_{\mathcal{E}}$ and fitting $(\hat{\psi}, \hat{\gamma})$, we can compute $\hat{S}_i \triangleq \hat{\mathbb{E}}[S_i \mid Y_{\mathcal{E}}]$ for all $i \in \mathcal{I}$. Then, we define our estimators for the distribution of performances and its corresponding quantile function 2.1 as

$$\hat{F}(x) \triangleq \frac{1}{I} \sum_{i \in \mathcal{I}} \mathbb{1}_{[\hat{S}_i, \infty)}(x) \quad \text{and} \quad \hat{Q}(p) \triangleq \inf\{x \in \mathbb{R} : \hat{F}(x) \geq p\}. \tag{3.5}$$

We name the procedure of obtaining $\hat{F}$ and $\hat{Q}$ as PromptEval and summarize it in Algorithm 1.

**Sampling $Y_{\mathcal{E}}$**   We have assumed $Y_{\mathcal{E}}$ is given so far. In practice, however, we need to choose $\mathcal{E}$, with $|\mathcal{E}| \leq B$ where $B \in \mathbb{N}$ is the budget, and then sample the entries $Y_{ij}$ for all $(i, j) \in \mathcal{E}$. One possible option is sampling $(i, j)$ without replacement from $\mathcal{I} \times \mathcal{J}$ giving the same sampling probability to all entries. This option is, however, suboptimal because of its high instability: with a high chance, there will be some prompt formats (or examples) with a very low number of evaluations while others will have many. A more stable solution is given by Algorithm 2, which balances the number of times each prompt format and examples are evaluated. Algorithm 2 can be seen as two-way stratified random sampling in which the number of examples observed for each prompt format is (roughly) the same and the number of prompt formats that observe each one of the examples is (roughly) the same.

# 4 Theoretical guarantees

In this section, we claim the consistency of the distribution and quantile estimators detailed in Algorithm 1 as $I, J \to \infty$. We prove a result for the case in which $f_\psi$ and $g_\gamma$ are linear/affine functions. Before we introduce our results we need to introduce some basic conditions. As an extra result, in Appendix I.1 we also show that our extended version of pIRT (3.4), X-pIRT, is uniformly consistent over all $i \in \mathcal{I}$, which can be useful beyond this work. We start by assuming that the covariates are uniformly bounded.

**Condition 4.1.** *There is a universal constant $c > 0$ such that $\sup_{i \in \mathcal{I}} \|x_i\|_2 , \sup_{j \in \mathcal{J}} \|z_j\|_2 < c$.*

The next condition requires the number of unseen examples to increase sufficiently fast as $I, J \to \infty$, which is a realistic condition under the low-budget setup. Weaker versions of this condition are possible; we adopt this one because it makes our proof simpler.

**Condition 4.2.** *Assume (i) $m = |\mathcal{J} \setminus \mathcal{J}_i|$ is the same for all $i$'s and grows to infinity and (ii) $\exp(\delta m)/I \to \infty$ as $I, J \to \infty$ for any $\delta > 0$.*

The third condition requires the model we work with to be correctly specified and the maximum likelihood estimator defined in (3.3) to be consistent as $I, J \to \infty$, *i.e.*, approach the true value. Evidently, $|\mathcal{E}|$ needs to grow to infinity as $I, J \to \infty$; nevertheless, it could be the case that $|\mathcal{E}|/(I \cdot J) \to 0$. When $f_\psi$ and $g_\gamma$ are linear/affine, the maximum likelihood procedure (3.3) is equivalent to fitting a logistic regression model and, in that case, the convergence of $(\hat{\psi}, \hat{\gamma})$ is well-studied and holds under mild conditions when the dimensions of the covariates are fixed; see, for example, Fahrmeir and Kaufmann [1985].

**Condition 4.3.** *The data point $Y_{ij}$ is sampled from a Bernoulli distribution with mean $\sigma(\psi_0^\top x_i - \gamma_0^\top z_j)$ for some true global parameter values $\psi_0$ and $\gamma_0$. Moreover, we assume that $(\hat{\psi}, \hat{\gamma}) \to (\psi_0, \gamma_0)$ in probability as $I, J \to \infty$.*

We now introduce the main result in Theorem 4.4, which shows the consistency of the distribution and quantile functions estimators introduced in Algorithm 1. See Appendix I for the proof.

**Theorem 4.4.** *Under conditions 4.1, 4.2, and 4.3, it is true that*

$$\left| \hat{Q}_\mathcal{I}(p) - Q_\mathcal{I}(p) \right| \to 0 \text{ in probability as } I, J \to \infty \text{ for any } p \in [0, 1],$$

*and that*

$$W_1(F, \hat{F}) \to 0 \text{ in probability as } I, J \to \infty,$$

*where $W_1(F, \hat{F})$ is the Wasserstein 1-distance between the distributions $F$ and $\hat{F}$.*

# 5 Assessing multi-prompt evaluation strategies

**General assessment** We assess the performance distribution and quantile function estimation methodology introduced in §3 in estimating the performance of LLMs and different prompt formats on data from three popular benchmarks. For a given LLM and a dataset, we consider two evaluation steps. First, we compare the full performance distribution with the estimated distribution, *i.e.*, in this case, all quantiles are considered. To compare the full performance distribution $F$ and its estimate $\hat{F}$, both defined in §3, we use the Wasserstein 1-distance which is equivalent to the average quantile estimation error in this case, *i.e.*,

$$W_1(F, \hat{F}) = \int_0^1 |Q(t) - \hat{Q}(t)| \mathrm{d}t = \frac{1}{I} \sum_{i=1}^I |S_{(i)} - \hat{S}_{(i)}|,$$

where $S_{(i)}$ (resp. $\hat{S}_{(i)}$) is the $i$-th smallest value in $\{S_i\}_{i \in \mathcal{I}}$ (resp. $\{\hat{\mathbb{E}}[S_i \mid Y_\mathcal{E}]\}_{i \in \mathcal{I}}$). Second, we estimate some quantiles of interest (*e.g.*, 5/25/50/75/95-th) for the performance distribution across prompt formats and compare them with the true quantiles, that is, for some $p \in [0, 1]$, we use $|Q(p) - \hat{Q}(p)|$ to measure the quality of our estimations.

**Data** We use data derived from three popular benchmarks: MMLU [Hendrycks et al., 2020], BIG-bench Hard (BBH) [Suzgun et al., 2022], and LMentry [Efrat et al., 2022]. In the following, we give more details about each one of the used datasets.

- MMLU is a multiple choice QA benchmark consisting of 57 subjects (tasks) comprising approximately 14k examples. We ran 15 different open-source LLMs (including different versions of Llama-3 [Meta, 2024], Mistral [Jiang et al., 2023], and Gemma [Gemma et al., 2024]) combined with 100 different prompt variations for each one of the MMLU tasks. We found that, within each one of the MMLU tasks, prompt templates can have great variability in their performances, making within-task analysis most suitable for assessing our method. More details and analysis of the collected data can be found in §7 and Appendix J.

- BIG-bench Hard (BBH) is a curated subset of BIG-bench [Srivastava et al., 2022], containing challenging tasks on which LLMs underperform the average human score. For BBH, we use the evaluation scores released by Mizrahi et al. [2023]. The evaluation data includes 11 open-source LLMs combined with a different number of prompt variations, ranging from 136 to 188 formats, for 15 tasks containing 100 examples each.

- LMentry consists of simple linguistic tasks designed to capture explainable and controllable linguistic phenomena. Like BBH, we use data generated by Mizrahi et al. [2023]. The authors made available the full evaluation data from 16 open-source LLMs combined with a different number of prompt variations, ranging from 226 to 259 formats, for 10 tasks containing from 26 to 100 examples each.

**Methods and baselines**   We consider different variations of the model presented in (3.2) coupled with Algorithm 1; for all variations, we use linear $f_\psi$ and $g_\gamma$. The most basic version of the model in (3.2) assumes $x_i$ and $z_j$ are one-hot encoded vectors, *i.e.*, vector of zeros with 1's on the entries $i$ and $j$, reverting the model to a Rasch model [Georg, 1960, Chen et al., 2023]. Despite its simplicity, we show that it can perform well in some cases. A more advanced instance of (3.2) assumes $x_i$ are either obtained using a sentence transformer [Reimers and Gurevych, 2019] to embed prompt templates or by extracting discrete covariates from the text, *e.g.*, as the presence of line breaks, colons *etc.*(see Appendix Table 2). An example of a prompt template for LMentry used by Mizrahi et al. [2023] is "*Can {category} be used to classify all the {words} provided? Respond with either "yes" or "no".*" Our method also allows using example covariates $z_j$, however, upon preliminary tests with sentence transformer we didn't observe improvements and chose to use one-hot-encoded vectors as in the basic Rasch model to represent examples. Next we detail the methods for obtaining the prompt covariates:

- *Prompt embeddings*. We embed prompt templates using a pre-trained sentence transformer variant [Karpukhin et al., 2020] and reduce their dimensionality to $d = 25$ using PCA. This is the most general solution that also works well in practice. We call it EmbPT.

- *Fine-tuned prompt embeddings*. Sentence transformers in general might not be most suitable for embedding prompt templates, thus we also consider fine-tuning BERT [Devlin et al., 2019] as an embedder. To do so, we use evaluation data for all examples and prompt formats from a subset of LLMs (these LLMs are excluded when assessing the quality of our estimators) and fine-tune `bert-base-uncased` to predict $Y_{ij}$ as in (3.3). We call this variation EmbFT and provide additional details in Appendix L. We acknowledge that obtaining such evaluation data for fine-tuning might be expensive, however, it might be justified in some applications if these embeddings provide sufficient savings for future LLM evaluations.

- *Discrete prompt covariates*. For BBH and LMentry, we coded a heuristic function that captures frequently occurring differences in common prompting templates. Examples of such covariates are the number of line breaks or the count of certain special characters (*e.g.*, dashes or colons). Each one of these covariates is encoded in $x_i$ for each one of the prompt templates $i \in \mathcal{I}$. A full list of the used heuristics is detailed in Appendix M. For MMLU, we adopted approach of [Sclar et al., 2023] to generate prompt variations via templates (see Algorithm 3), which also provides a natural way to construct the covariates, *e.g.*, the presence of dashes or colons.

To the best of our knowledge, the methods introduced in §3 are the first ones handling the problem of efficient evaluation of performance *distribution* of LLMs across multiple prompts. Thus, we compare different variations of our method with one natural baseline ("avg") which estimates $S_i$ by simply averaging $Y_{ij}$, that is, using the estimates $\hat{S}_i^{\text{avg}} = \frac{1}{|\mathcal{J}_i|} \sum_{j \in \mathcal{J}_i} Y_{ij}$. The estimates for the distribution and quantile function are then obtained by computing the function in (3.5) using $\hat{S}_i^{\text{avg}}$ instead of $\hat{S}_i$. To make comparisons fair, we sample the data using Algorithm 2 for all methods and the baseline.

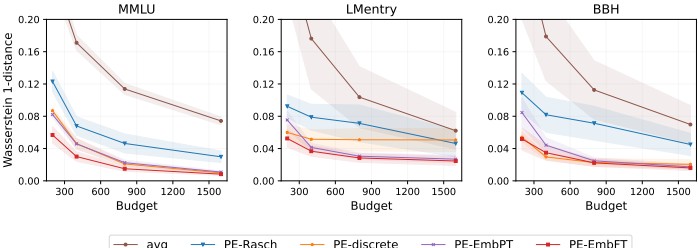

Figure 2: Performance distribution estimation errors measured with Wasserstein-1 distance on three benchmarks.

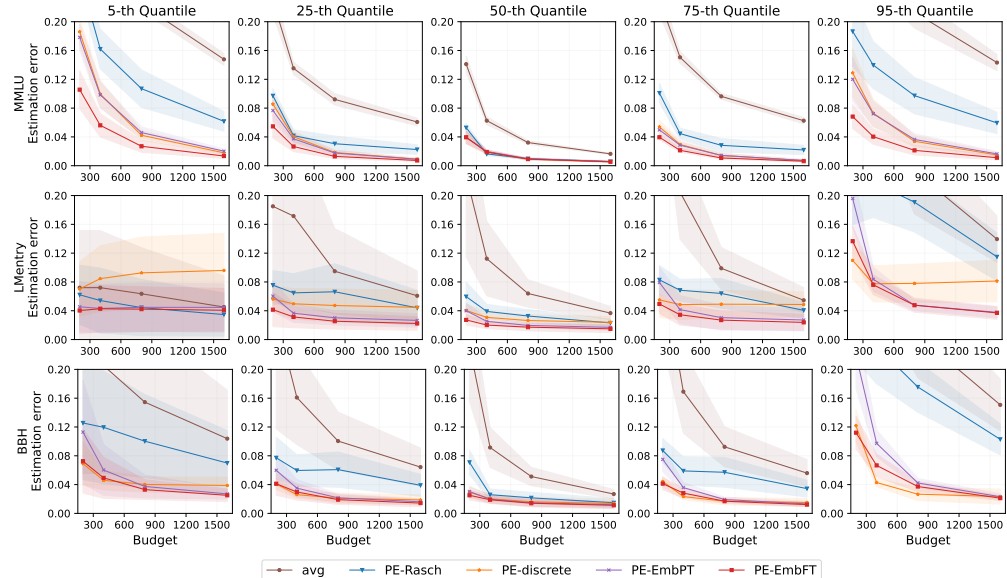

Figure 3: Performance quantile estimation errors for varying quantiles (columns) and benchmarks (rows).

**Key results**    We investigate the effectiveness of the different variations of PromptEval (PE) against the "avg" baseline strategy in quantile estimation and overall performance distribution estimation across prompt templates. In total, we consider five variations of PromptEval: (i) PE-Rasch (model in (3.2) is a Rach model), (ii) PE-discrete (discrete covariates are used for prompt templates), (iii) PE-EmbPT (pre-trained LLM embeddings are used for prompt templates), and (iv) PE-EmbFT (fine-tuned LLM embeddings are used for prompt templates). Within each one of the benchmarks, we conduct a different experiment for each one of the tasks, LLMs, and 5 random seeds used when sampling $Y_{\mathcal{E}}$. We report the average estimation error across tasks, LLMs, and seeds, while the error bars are for the average estimation errors across LLMs. We collect results for four different numbers of total evaluations, where $|\mathcal{E}| \in \{200, 400, 800, 1600\}$. To make our results more tangible, 200 evaluations are equivalent, on average, to $1.15\%$ to the total number of evaluations on BBH, $0.88\%$ to the total number of evaluations on LMentry, and $0.81\%$ to the total number of evaluations on MMLU.

- *Distribution estimation.* Our results for distribution estimation can be seen in Figure 2. We see that, in general, all variations of PromptEval, including its simplest version (*PE-Rasch*), can do much better in distribution estimation when compared to the baseline. Among our methods, the ones that use covariates are the best ones.

- *Quantile estimation.* Our results for quantile estimation are presented in Figure 3. As before, even the simplest version of our method (*PE-Rasch*) does much better than the considered baseline. For all the other variations of PromptEval, estimating extreme quantiles is usually hard and needs more evaluations, while more central ones (*e.g.*, median) can be accurately estimated with 200 evaluations, providing more than 100x compute saving in most cases. Regarding the different variations of PromptEval, we found that the pre-trained embeddings are robust across benchmarks, while the discrete covariates could not do well on LMentry data. Using covariates obtained via fine-tuning the BERT model provides some further improvements, for example, for extreme quantiles and small evaluation budget settings on MMLU. However, fine-tuning requires collecting

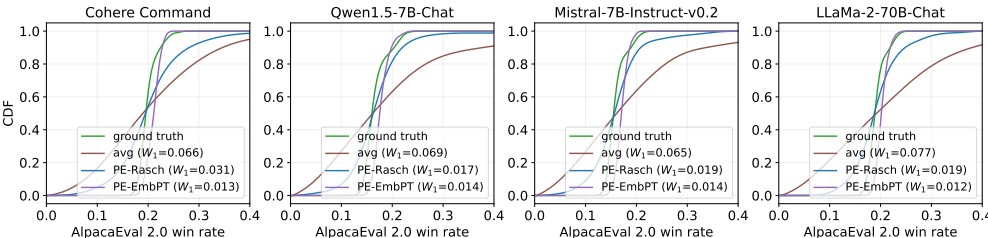

Figure 4: Estimating LLM-as-a-judge distribution of scores for 100 prompt variations given to the judge.

large amounts of evaluation data and in most cases, we anticipate that it would be more practical to use PromptEval with pre-trained embedder and moderate evaluation budget instead.

# 6 Further applications of PromptEval

## 6.1 Estimating the distribution of scores for the LLM-as-a-judge framework

In this subsection, we explore the concept of LLM-as-a-judge using the AlpacaEval 2.0 [Li et al., 2023] benchmark. Specifically, we generate 100 prompt templates[5] to present to the judge, GPT-4o-mini [OpenAI, 2024], allowing us to assess how sensitive model evaluation is to different evaluation prompts. We evaluated the performance of four LLMs with similar capabilities (Cohere Command[6], Qwen1.5-7B-Chat [Team, 2024], Mistral-7B-Instruct-v0.2 [Jiang et al., 2023], LLaMa-2-70B-Chat [Touvron et al., 2023]) using only $\approx 2\%$ of the total evaluations (1.6k/80.5k). In contrast to the previous experiments, we do not make changes in the prompt templates given to the evaluated LLMs when giving an instruction. To fit PromptEval, we binarize AlpacaEval 2.0 instance scores imposing a threshold at $1/2$, but we do not binarize the responses at test time. Figure 4 shows that the different variations of PromptEval can obtain a much lower Wasserstein loss ($W_1$) when compared with the baseline "avg". In Appendix G, we provide additional plots for this experiment. Specifically, Figure 11 presents the score distribution histograms for the four models under consideration, while Figure 12 illustrates how certain prompt templates consistently lead the judge to assign higher (or lower) scores across models. Despite this pattern, we observe that the ranking of the four LLMs changes in 36% of the prompt templates.

## 6.2 Best-prompt identification

The best-prompt identification task [Shi et al., 2024] is to find the best prompt from a set of fixed templates, *i.e.*, the one that gives the best performance for a task at hand. Shi et al. [2024] propose framing this problem as a bandit problem and using a linear model or an MLP to predict the performance of each prompt template. To apply PromptEval in this setting we use our model (3.2) and X-pIRT to estimate how good each template is coupled with sequential elimination algorithm [Azizi et al., 2021] (as in Shi et al. [2024]) to select prompt-example pairs for evaluation in each round. In Figure 5 we compare our PE to the baseline TRIPLE-GSE [Shi et al., 2024] with a logistic regression performance predictor and the same three types of covariates (PE-OneHot corresponds to PE-Rasch

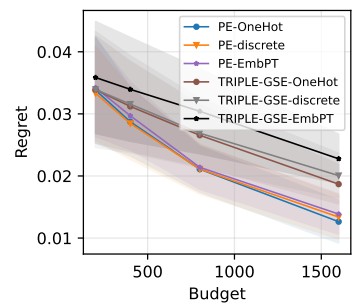

Figure 5: Best-prompt identification.

in previous experiments). For all covariate choices, we show that using PromptEval for best-prompt identification results in lower regret, *i.e.*, the performance of the best template minus the performance of the chosen template. We include the full results for other benchmarks and also apply TRIPLE-GSE with an MLP in Appendix H.

---

[5]We generate 10k variations using ChatGPT and undersample to 100 deleting prompts that are too similar to each other. You can see our code here.

[6]https://docs.cohere.com/v2/docs/command-beta

## 7 Analysis of prompt sensitivity on MMLU

Prior work reports strong sensitivity of LLMs to spurious prompt template changes (see Section 1.1). For example, Sclar et al. [2023] observe performance changes of up to 80% for Natural Instructions tasks [Wang et al., 2022] due to template changes. Despite its popularity, no such analysis exists for the MMLU dataset to date. We here provide an in-depth analysis of MMLU prompt sensitivity.

**Performance spread** When averaged across subjects, we observe relatively small performance spreads per LLM compared to other datasets in the literature (see Figure 16 in the Appendix K). For example, we can consistently identify `Llama-3-70B-Instruct` as the best performing model, independent of the prompt template. On the other hand, scores within individual subjects are highly inconsistent. Figure 6 shows the distribution of prompt spreads (max-min acc.) across subjects per LLM. Most LLMs demonstrate a significant average spread of around 10% at the subject level.

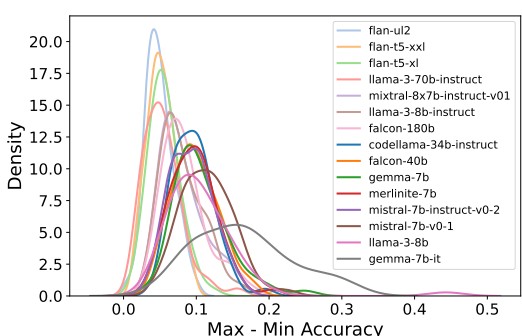

Figure 6: Accuracy spread across 57 subjects.

**Template consistency** In practice, having consistently performing templates is highly relevant *within a single LLM* or *across LLMs* for the same subject. To evaluate the template consistency, we rank template performances either across subjects or across LLMs to then calculate the agreement across those rankings using Kendall's $W$ [Kendall and Smith, 1939, inspired by Mizrahi et al. 2023].

Within LLMs, we observe that `Gemma-7B-it` has a notably higher Kendall's $W$ of 0.45 than any other model, meaning a fixed set of prompts performs best across many subjects (for full results, see Table 1 in the Appendix). Across LLMs, we do not observe high correlations within any of the subjects (see Figure 17 in Appendix K). Hence, similar to previous findings [*e.g.* Sclar et al., 2023], we do not identify any coherent template preferences across LLMs (for detailed results, see Appendix K).

## 8 Conclusion

PromptEval enables a more comprehensive evaluation of LLMs. We hope that comparing distributions or quantiles across many prompt variants will enable more robust leaderboards and address the common concern of comparing LLMs with a single pre-defined prompt. Prior to our work, a major limitation of such evaluation was its cost. We demonstrated empirically across several popular benchmarks that our method can produce accurate performance distribution and quantile estimates at the cost of 2-4 single-prompt evaluations, out of hundreds possible. However, several questions remain: how to decide on the set of prompts for evaluation and how to best utilize our distribution estimates for comparison in various contexts. For the former, we utilized suggestions from prior work [Mizrahi et al., 2023, Sclar et al., 2023] and for the latter, we primarily focused on quantiles as well-established robust performance measures.

Besides evaluation, another common problem in practice is finding the best prompt for a given task. Our method can be applied in this setting when there is a pre-defined set of candidate prompts (Figure 5). However, several recent works [Prasad et al., 2023, Yang et al., 2023, Li and Wu, 2023, Ye et al., 2023a] demonstrate the benefits of dynamically generating new prompt candidates. For example, Prasad et al. [2023] propose an evolutionary algorithm that creates new prompts based on the ones that performed well at an earlier iteration. Extending PromptEval to accommodate an evolving set of prompt candidates is an interesting future work direction.

We comment on the limitations of our work in Appendix A.

## 9 Acknowledgements

This paper is based upon work supported by the National Science Foundation (NSF) under grants no. 2027737 and 2113373.

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

# A    Limitations

While our method offers a more reliable and flexible evaluation, it relies on using multiple prompts. As a result, if selecting a single prompt was a challenge in earlier benchmarks, determining the appropriate set of prompt templates now becomes the key challenge. Although methods have been proposed for generating and diversifying multiple prompts [Mizrahi et al., 2023], and the selection of individual prompts becomes less critical when many are used, this remains a limitation for future work to address.

Another limitation is that we do not focus on prompt engineering or attempt to solve this issue. While addressing prompt engineering would be a significant contribution to the field, our approach assumes a predefined set of prompts and focuses solely on evaluation or optimization within that pool of prompts. This is a practical and widely used setting but does not address the broader prompt engineering challenge.

# B    Adapting the correctness model for bounded $Y_{ij}$

There might be situations in LLM evaluation in which $Y_{ij} \notin \{0, 1\}$ but $Y_{ij} \in [0, 1]$. For example, in AlpacaEval 2.0 [Li et al., 2023], the response variable is bounded and can be translated to the interval $[0, 1]$. Also, some scenarios of HELM [Liang et al., 2022] and the Open LLM Leaderboard [Beeching et al., 2023] have scores in $[0, 1]$. One possible fix is changing the model for $Y_{ij}$. For example, if $Y_{ij}$ are continuous, the Beta model would be appropriate. Another possibility that offers a more immediate fix is binarizing $Y_{ij}$ as proposed by Maia Polo et al. [2024]. That is, using a training set containing correctness data from $L$ LLMs, we could find a constant $c$ such that $\sum_{i,j,l} Y_{ijl} \approx \sum_{i,j,l} \mathbb{1}[Y_{ijl} \geq c]$, where the index $l$ represents each LLM in the training set. Then, we define $\tilde{Y}_{ij} \triangleq \mathbb{1}[Y_{ij} \geq c]$ and work with this newly created variable.

# C    Comments on the computational complexity of PromptEval

Consider the case of our experiments in which prompts are represented by embeddings of fixed size, examples are represented by one-hot encodings, and our model is given by logistic regression. Because the dimension of the embeddings does not depend on the number of prompt variations, the number of samples and variables used to fit our model does not vary with the number of prompt variations. Then, computational costs are constant with respect to the number of prompt templates. On the other hand, the number of variables (and consequently samples, to make estimation possible) should increase linearly with the number of examples, which are usually hundreds or a few thousand. Thus, this should not be a problem in most practical cases.

# D    Computing resources

All experiments were conducted using a virtual machine with 32 cores. The results for each benchmark separately can be obtained within 3-6 hours.

For fine-tuning BERT embeddings, we employ multiple NVIDIA A30 GPUs with 24 GB vRAM, requiring 70 hours of training and an additional approximately 350 hours for hyperparameter search. Fine-tuning can be conducted on GPUs with smaller capacities.

# E    Estimation errors by task

In Figures 7, 8, and 9, we analyze the Wasserstein 1-distance per task for each benchmark when using the method PE-EmbPT, a robust and versatile variation of PromptEval. The results show that for BBH and LMentry, the estimation errors (Wasserstein 1-distance) are more uniform across tasks compared to MMLU, where some tasks exhibit higher estimation errors. This discrepancy occurs because all tasks in BBH and LMentry have the same number of examples, whereas tasks in MMLU, particularly those with higher estimation errors, have a significantly larger number of examples when compared to the others. In those cases, a larger number of evaluations is recommended.

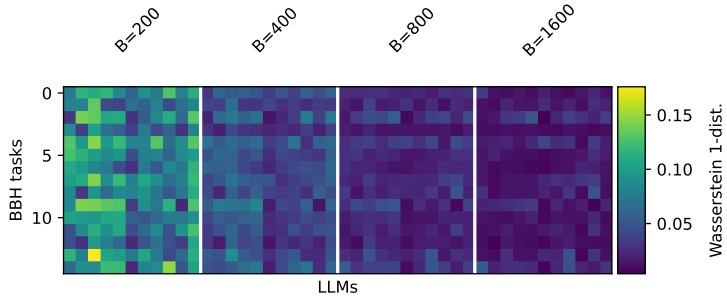

Figure 7: Estimation error for the BBH tasks.

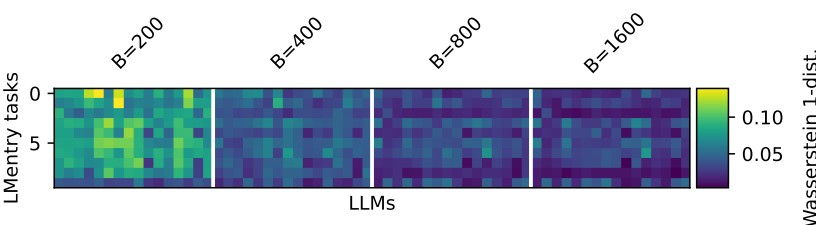

Figure 8: Estimation error for the LMentry tasks.

## F  The influence of the number of prompts in PromptEval performance

We repeat the main experiment in the paper (randomly) cutting the number of prompt templates by a factor of 5. This means we use only 20 prompt variations in MMLU, for example. In summary, PromptEval still does well, beating the baseline. However, the gap between PromptEval and the baseline has shortened due to fewer variations. This fact highlights that the bigger the number of templates, the more useful PromptEval can be relative to the baseline. The results are depicted in Figure 10.

## G  Extra plots for the LLM-as-a-judge experiment

In Figure 11, we show the performance distribution histograms for the four considered models across prompt templates. Figure 12 shows that prompt templates consistently lead the judge to assign higher (or lower) scores across models; in this plot, we normalize the scores within rows so 0 is assigned to the lowest score and 1 is assigned to the maximum score (brighter colors denote higher scores).

## H  Extra results for best-prompt identification

In Figures 13, 14, and 15, we can see the full results for MMLU, BBH, and LMentry. For all benchmarks, we can see that within each triple "PE", "TRIPLE-GSE", "TRIPLE-MLP-GSE", the "PE" version always has some advantage with a lower regret.

The tuning and fitting process of the Multi-Layer Perceptron (MLP) classifier involves setting up a pipeline that includes feature scaling and the MLP classifier itself, which has 30 neurons in its hidden layer. This process begins by defining a range of values for critical hyperparameters: the l2 regularization strength is tested over the range from 0.001 to 10, and the initial learning rate is tested over the range from 0.001 to 0.1. These values are systematically tested through cross-validation to determine the optimal combination. During this phase, cross-validation ensures that the model is evaluated on different subsets of the data to prevent overfitting and to ensure robust performance. Once the best hyperparameters are identified, the final model is trained on the entire dataset using these optimal settings, resulting in a well-tuned MLP classifier ready for deployment.

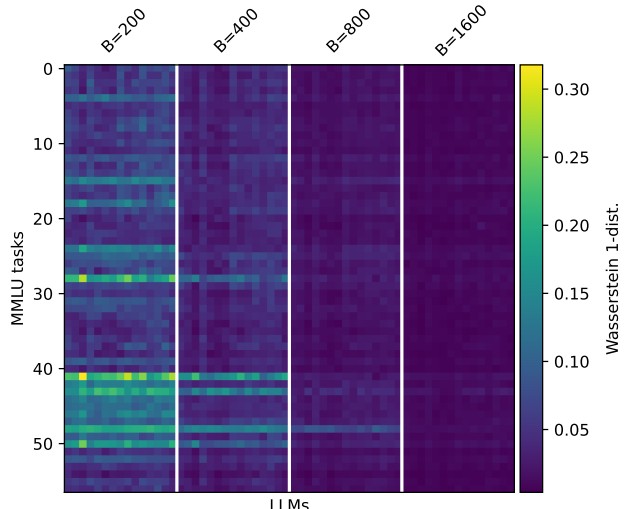

Figure 9: Estimation error for the MMLU tasks.

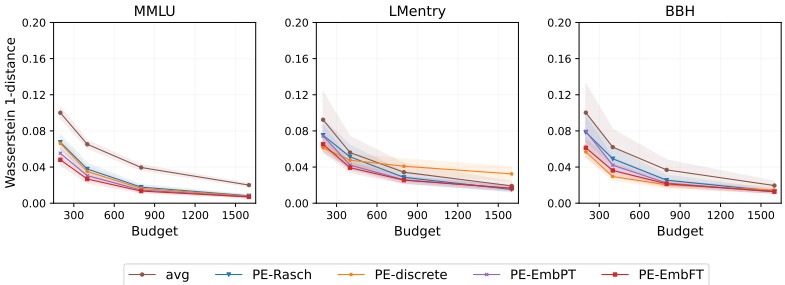

Figure 10: Performance estimation quality when we keep only 20% of the prompts variations for each task. In summary, PromptEval still does beat the baseline. However, the gap between PromptEval and the baseline has shortened due to fewer variations.

# I Theoretical results

## I.1 Consistency of X-pIRT

In Theorem I.1, we claim that the X-pIRT estimator is uniformly consistent over all $i \in \mathcal{I}$.

**Theorem I.1.** *Under conditions 4.1, 4.2, and 4.3, it is true that*

$$\sup_{i \in \mathcal{I}} \left| \hat{\mathbb{E}}[S_i \mid Y_{\mathcal{S}}] - S_i \right| \to 0 \text{ in probability as } I, J \to \infty.$$

A direct consequence of Theorem I.1 is that

$$\left| \frac{1}{I} \sum_{i \in \mathcal{I}} \hat{\mathbb{E}}[S_i \mid Y_{\mathcal{S}}] - \frac{1}{I} \sum_{i \in \mathcal{I}} S_i \right| \leq \frac{1}{I} \sum_{i \in \mathcal{I}} \left| \hat{\mathbb{E}}[S_i \mid Y_{\mathcal{S}}] - S_i \right| \leq \sup_{i \in \mathcal{I}} \left| \hat{\mathbb{E}}[S_i \mid Y_{\mathcal{S}}] - S_i \right| \to 0$$

in probability as $I, J \to \infty$. This means that the mean of predicted performances is also consistent if a practitioner wants to use it as a summary statistic.

The proof of Theorem I.1 is embedded in the proof of Theorem 4.4.

## I.2 Proof of Theorem 4.4

For the following results, we denote $\psi^\top x_i$ as $\theta_i$ and $\gamma^\top z_j$ as $\beta_j$, and $\hat{\psi}^\top x_i$ as $\hat{\theta}_i$ and $\hat{\gamma}^\top z_j$ as $\hat{\beta}_j$. Moreover, if a sequence random variables $(X_n)$ converge to 0 in distribution, we denote $X_n = o_P(1)$.

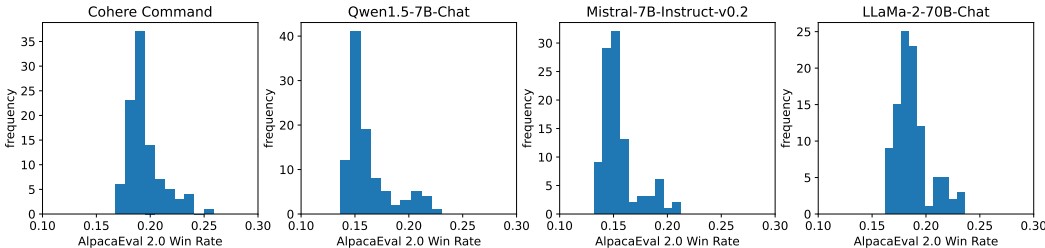

Figure 11: Performance distribution histograms for the four considered models across prompt templates.

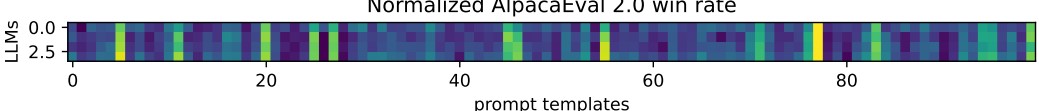

Figure 12: Prompt templates consistently lead the judge to assign higher (or lower) scores across models..

**Lemma I.2.** *Under Conditions 4.1 and 4.3, we have that* $\sup_{i \in \mathcal{I}} |\hat{\theta}_i - \theta_i| = o_P(1)$ *and* $\sup_{j \in \mathcal{J}} |\hat{\beta}_j - \beta_j| = o_P(1)$ *as* $I, J \to \infty$.

*Proof.* We prove that $\sup_{i \in \mathcal{I}} |\hat{\theta}_i - \theta_i| = o_P(1)$. The second statement is obtained in the same way.

See that

$$\sup_{i \in \mathcal{I}} |\hat{\theta}_i - \theta_i| = \sup_{x_i} |(\hat{\psi} - \psi)^\top x_i| \leq \sup_{x_i} \left\| \hat{\psi} - \psi \right\|_2 \|x_i\|_2 \leq c \left\| \hat{\psi} - \psi \right\|_2 = o_P(1)$$

as $I, J \to \infty$. Where the first inequality is obtained using the Cauchy–Schwarz inequality, the second is obtained using Condition 4.1, and the last equality is a consequence of Condition 4.3 and the continuous mapping theorem [Resnick, 2019]. □

**Lemma I.3.** *Under Conditions 4.1 and 4.3, it is true that*

$$\sup_{i \in \mathcal{I}} \left| \hat{\mathbb{E}}[S_i \mid Y_\mathcal{E}] - \mathbb{E}[S_i \mid Y_\mathcal{E}] \right| = o_P(1) \text{ as } I, J \to \infty.$$

*Proof.* See that

$$
\begin{aligned}
\sup_{i \in \mathcal{I}} \left| \hat{\mathbb{E}}[S_i \mid Y_\mathcal{E}] - \mathbb{E}[S_i \mid Y_\mathcal{E}] \right| &= \sup_{i \in \mathcal{I}} \frac{1 - \lambda_i}{J - |\mathcal{J}_i|} \left| \sum_{j \notin \mathcal{J}_i} \sigma(\hat{\theta}_i - \hat{\beta}_j) - \sigma(\theta_i - \beta_j) \right| \\
&\leq \sup_{i \in \mathcal{I}} \frac{1 - \lambda_i}{J - |\mathcal{J}_i|} \sum_{j \notin \mathcal{J}_i} \left| \sigma(\hat{\theta}_i - \hat{\beta}_j) - \sigma(\theta_i - \beta_j) \right| \\
&\leq \sup_{i \in \mathcal{I}} \frac{1 - \lambda_i}{4(J - |\mathcal{J}_i|)} \sum_{j \notin \mathcal{J}_i} \left| \hat{\theta}_i - \hat{\beta}_j - \theta_i + \beta_j \right| \\
&\leq \frac{1 - \inf_i \lambda_i}{4} \left( \sup_j |\hat{\theta}_i - \theta_i| + \sup_j |\hat{\beta}_j - \beta_j| \right) \\
&\leq \frac{1}{4} \left( \sup_i |\hat{\theta}_i - \theta_i| + \sup_j |\hat{\beta}_j - \beta_j| \right) \\
&= o_P(1)
\end{aligned}
$$

where the third step is justified by the fact that $\sigma$ is $1/4$-Lipschitz and the last step is justified by Lemma I.2. □

**Lemma I.4.** *Under Condition 4.2, it is true that*

$$\sup_{i \in \mathcal{I}} |\mathbb{E}[S_i \mid Y_\mathcal{E}] - S_i| = o_P(1) \text{ as } I, J \to \infty.$$

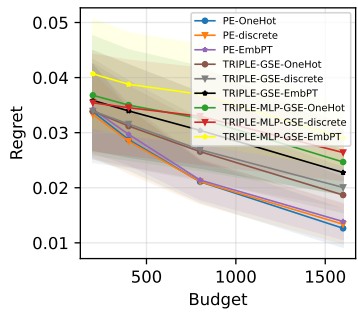

Figure 13: Best-prompt identification for MMLU

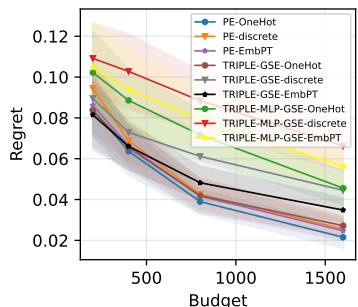

Figure 14: Best-prompt identification for BBH

*Proof.* For an arbitrary $\epsilon > 0$, see that

$$
\mathbb{P}\left(\sup_{i\in\mathcal{I}}|\mathbb{E}[S_i \mid Y_{\mathcal{E}}] - S_i| \geq \epsilon\right) = \mathbb{P}\left(\bigcup_{i\in\mathcal{I}}\{|\mathbb{E}[S_i \mid Y_{\mathcal{E}}] - S_i| \geq \epsilon\}\right)
$$

$$
\leq \sum_{i\in\mathcal{I}}\mathbb{P}\left(|\mathbb{E}[S_i \mid Y_{\mathcal{E}}] - S_i| \geq \epsilon\right)
$$

$$
= \sum_{i\in\mathcal{I}}\mathbb{P}\left(\left|\frac{\lambda_i}{|\mathcal{J}_i|}\sum_{j\in\mathcal{J}_i}Y_{ij} + \frac{1-\lambda_i}{|\mathcal{J}\setminus\mathcal{J}_i|}\sum_{j\notin\mathcal{J}_i}\sigma(\theta_i - \beta_j) - \frac{1}{J}\sum_{j\in\mathcal{J}}Y_{ij}\right| \geq \epsilon\right)
$$

$$
= \sum_{i\in\mathcal{I}}\mathbb{P}\left(\left|(1-\lambda_i)\frac{1}{|\mathcal{J}\setminus\mathcal{J}_i|}\sum_{j\notin\mathcal{J}_i}Z_{ij}\right| \geq \epsilon\right)
$$

$$
\leq \sum_{i\in\mathcal{I}}\mathbb{P}\left(\left|\frac{1}{m}\sum_{j\notin\mathcal{J}_i}Z_{ij}\right| \geq \epsilon\right)
$$

where $Z_{ij} \triangleq Y_{ij} - \sigma(\theta_i - \beta_j)$. Consequently, $|Z_{ij}| \leq 1$ and $\mathbb{E}[Z_{ij}] = 0$. Applying Hoeffding's inequality, we obtain

$$
\mathbb{P}\left(\sup_{i\in\mathcal{I}}|\mathbb{E}[S_i \mid Y_{\mathcal{E}}] - S_i| \geq \epsilon\right) \leq 2I\exp\left(-2m\epsilon^2\right)
$$

$$
= 2\exp\left(\log I - 2\epsilon^2 m\right)
$$

$$
= 2\exp\left(-\log(\exp(2\epsilon^2 m)/I)\right)
$$

$$
\to 0
$$

$\square$

**Lemma I.5.** *Let $a_1, \cdots, a_n$ and $b_1, \cdots, b_n$ be two lists of real numbers and let $a_i$ and $b_j$ be the p-lower quantiles of those lists. Admit that there are $m_1$ a's lower than $a_i$, $m_2$ a's equal to $a_i$ (besides*

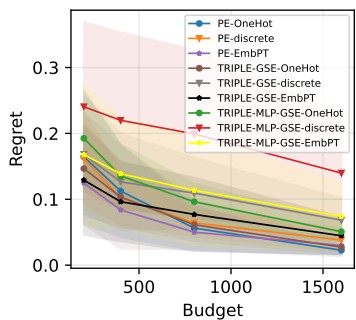

Figure 15: Best-prompt identification for LMentry

$a_i$ itself), and $m_3$ a's greater than $a_i$. Then, there are at least $m_1 + 1$ b's lower or equal to $b_j$ and at least $m_3 + 1$ b's greater or equal to $b_j$.

*Proof.* If $a_i$ is the $p$-lower quantile of $\mathcal{A} = \{a_1, \cdots, a_n\}$, then by definition $a_i$ is the lowest value in $\mathcal{A}$ such that

$$\underbrace{|\{a \in \mathcal{A} : a_i = a\}|}_{m_2+1} + \underbrace{|\{a \in \mathcal{A} : a_i > a\}|}_{m_1} \geq p \cdot n$$

Because $a_i$ is the lowest value in $\mathcal{A}$ to achieve that, then $m_1 < p \cdot n$. This implies that there are at least $m_1 + 1$ values in $\mathcal{B} = \{b_1, \cdots, b_n\}$ lower or equal to $b_j$ as it is the $p$-lower quantile of $\mathcal{B}$.

Finally, because $m_1 + m_2 + 1 \geq p \cdot n$, we know that $\mathcal{B}$ cannot have more than $m_1 + m_2$ values strictly lower than $b_j$, otherwise the $p$-lower quantile of $\mathcal{B}$ could not be $b_j$ but some of those values. Therefore, $\mathcal{B}$ have at least $m_3 + 1$ values greater or equal to $b_j$. $\square$

**Lemma I.6.** *Let $\hat{i}$ and $i^*$ be indices in $\mathcal{I}$ such that $\hat{Q}(p) = \hat{\mathbb{E}}[S_{\hat{i}} \mid Y_S]$ and $Q(p) = S_{i^*}$ for an arbitrary fixed $p \in [0, 1]$. Under $\sup_{i \in \mathcal{I}} |\hat{\mathbb{E}}[S_i \mid Y_{\mathcal{E}}] - S_i| \leq \epsilon$ for an arbitrary $\epsilon > 0$, if $|\hat{\mathbb{E}}[S_{i'} \mid Y_{\mathcal{E}}] - S_{i^*}| > 2\epsilon$, for some $i' \in \mathcal{I}$, then $\hat{i} \neq i'$. Consequently, if $\hat{i} = i'$ then $|\hat{\mathbb{E}}[S_{i'} \mid Y_{\mathcal{E}}] - S_{i^*}]| \leq 2\epsilon$ under $\sup_{i \in \mathcal{I}} |\hat{\mathbb{E}}[S_i \mid Y_{\mathcal{E}}] - S_i| \leq \epsilon$.*

*Proof.* Define $\mathcal{A} \triangleq \{S_1, \cdots, S_i\}$ and assume that there are $M_1$ values in $\mathcal{A}$ lower than $S_{i^*}$, $M_2$ values equal to $S_{i^*}$ (besides $S_{i^*}$ itself), and $M_3$ values greater than $S_{i^*}$. If $|S_{i'} - S_{i^*}| > 2\epsilon$, for a certain index $i'$, there are two possibilities: (i) $S_{i'} + 2\epsilon < S_{i^*}$ or (ii) $S_{i^*} + 2\epsilon < S_{i'}$. Under the event $\sup_{i \in \mathcal{I}} |\hat{\mathbb{E}}[S_i \mid Y_{\mathcal{E}}] - S_i| \leq \epsilon$, we have:

- If (i) holds, then there is at least $M_2 + M_3 + 1$ values of $\hat{\mathbb{E}}[S_i \mid Y_{\mathcal{E}}]$'s such that $\hat{\mathbb{E}}[S_{i'} \mid Y_{\mathcal{E}}] < \hat{\mathbb{E}}[S_i \mid Y_{\mathcal{E}}]$ (including $\hat{\mathbb{E}}[S_{i^*} \mid Y_{\mathcal{E}}]$). This implies that at most $M_1$ values of $\hat{\mathbb{E}}[S_i \mid Y_{\mathcal{E}}]$'s will be less or equal $\hat{\mathbb{E}}[S_{i'} \mid Y_{\mathcal{E}}]$. By Lemma I.5, we know that $j' \neq \hat{j}$.

- If (ii) holds, then there is at least $M_1 + M_2 + 1$ values of $\hat{\mathbb{E}}[S_i \mid Y_{\mathcal{E}}]$'s such that $\hat{\mathbb{E}}[S_{i'} \mid Y_{\mathcal{E}}] > \hat{\mathbb{E}}[S_i \mid Y_{\mathcal{E}}]$ (including $\hat{\mathbb{E}}[S_{i^*} \mid Y_{\mathcal{E}}]$). This implies that at most $M_3$ values of $\hat{\mathbb{E}}[S_i \mid Y_{\mathcal{E}}]$'s will be greater or equal $\hat{\mathbb{E}}[S_{i'} \mid Y_{\mathcal{E}}]$. By Lemma I.5, we know that $j' \neq \hat{j}$.

This means that under $\sup_{i \in \mathcal{I}} |\hat{\mathbb{E}}[S_i \mid Y_{\mathcal{E}}] - S_i| \leq \epsilon$ for an arbitrary $\epsilon > 0$, if $|\hat{\mathbb{E}}[S_{i'} \mid Y_{\mathcal{E}}] - S_{i^*}| > 2\epsilon$, for some $i' \in \mathcal{I}$, then $\hat{i} \neq i'$. $\square$

***Proof of Theorem 4.4 (Part 1).*** Let $\hat{i}$ and $i^*$ be indices in $\mathcal{I}$ such that $\hat{Q}(p) = \hat{\mathbb{E}}[S_{\hat{i}} \mid Y_S]$ and $Q(p) = S_{i^*}$. Notice that Lemma I.6 guarantees that $\sup_{i \in \mathcal{I}} |\hat{\mathbb{E}}[S_i \mid Y_S] - S_i| \leq \epsilon$ implies $|\hat{\mathbb{E}}[S_{\hat{i}} \mid Y_S] - S_{i^*}| \leq 2\epsilon$, for an arbitrary $\epsilon > 0$. Consequently,

$$\mathbb{P}\left(|\hat{\mathbb{E}}[S_{\hat{i}} \mid Y_S] - S_{i^*}| \leq 2\epsilon\right) \geq \mathbb{P}\left(\sup_{i \in \mathcal{I}} |\hat{\mathbb{E}}[S_i \mid Y_S] - S_i| \leq \epsilon\right) = 1 + o(1)$$

because

$$\sup_{i \in \mathcal{I}} \left| \hat{\mathbb{E}}[S_i \mid Y_{\mathcal{E}}] - S_i \right| \leq \sup_{i \in \mathcal{I}} \left| \hat{\mathbb{E}}[S_i \mid Y_{\mathcal{E}}] - \mathbb{E}[S_i \mid Y_{\mathcal{E}}] \right| + \sup_{i \in \mathcal{I}} \left| \mathbb{E}[S_i \mid Y_{\mathcal{E}}] - S_i \right| = o_P(1) \text{ as } I, J \to \infty$$

holds by lemmas I.3 and I.4. Because $\epsilon > 0$ is arbitrary, we have that $|\hat{Q}(p) - Q(p)| = o_P(1)$.

$\square$

***Proof of Theorem 4.4 (Part 2).*** We start this proof by showing that

$$|\hat{Q}(U) - Q(U)| = o_P(1)$$

with $U \sim \text{Unif}[0,1]$ independent of $\hat{Q}$ and $Q$.

For an arbitrary $\epsilon > 0$, see that

$$\lim_{I,J \to \infty} \mathbb{P}(|\hat{Q}(U) - Q(U)| > \epsilon) = \lim_{I,J \to \infty} \mathbb{E}\left[ \mathbb{P}(|\hat{Q}(U) - Q(U)| > \epsilon \mid U) \right]$$
$$= \mathbb{E}\left[ \lim_{I,J \to \infty} \mathbb{P}(|\hat{Q}(U) - Q(U)| > \epsilon \mid U) \right]$$
$$= 0$$

where the second equality is justified by the Dominated Convergence Theorem [Resnick, 2019] and the last one is justified by $|\hat{Q}(p) - Q(p)| = o_P(1)$. Now, we see that

$$\lim_{I,J \to \infty} \mathbb{E}[W_1(F, \hat{F})] = \lim_{I,J \to \infty} \mathbb{E}\left[ \int_0^1 |Q(t) - \hat{Q}(t)| \mathrm{d}t \right]$$
$$= \lim_{I,J \to \infty} \mathbb{E}\left[ |\hat{Q}(U) - Q(U)| \right]$$
$$= 0$$

where the last step is justified by Fubini's Theorem [Resnick, 2019], $|\hat{Q}(U) - Q(U)| = o_P(1)$, and the Lebesgue Dominated Convergence Theorem [Resnick, 2019]. For an arbitrary $\epsilon > 0$ and applying Markov's inequality, we get

$$\lim_{I,J \to \infty} \mathbb{P}(W_1(F, \hat{F}) > \epsilon) \leq \frac{1}{\epsilon} \lim_{I,J \to \infty} \mathbb{E}[W_1(F, \hat{F})] = 0$$

$\square$

## J  Details MMLU data

Algorithm 3 for automatically generating templates can be seen as a graph traversal of a template graph, whose nodes are defined by which features they have: a separator $SEP$, a space $SPA$, and an operator $OP$. By traversing this graph, we can collect unique templates that can used in the evaluation of LLMs on tasks.

---
**Algorithm 3:** TemplateGeneration
---
1 **Input:** Base prompt template features: Separator $SEP$, Space $SPA$, Operator $OP$.

2 **Output:** Prompt templates.

3 From template agenda, pop a template. Swap $SEP$ with another $SEP$, add to templates. Swap $SPA$ with another $SPA$, add to templates. Swap $OP$ with another $OP$, add to templates. Add the generated templates to the agenda.

4 **return** generated templates.
---

Next, we utilize the unitxt [Bandel et al., 2024] preprocessing library to build custom datasets with the generated templates. Standardized and accurate evaluation is then carried out via the LM-Eval-Harness [Gao et al., 2023] evaluation library.

# K Details MMLU spread analysis

Figure 16 depicts the performance of LLMs on the whole MMLU.

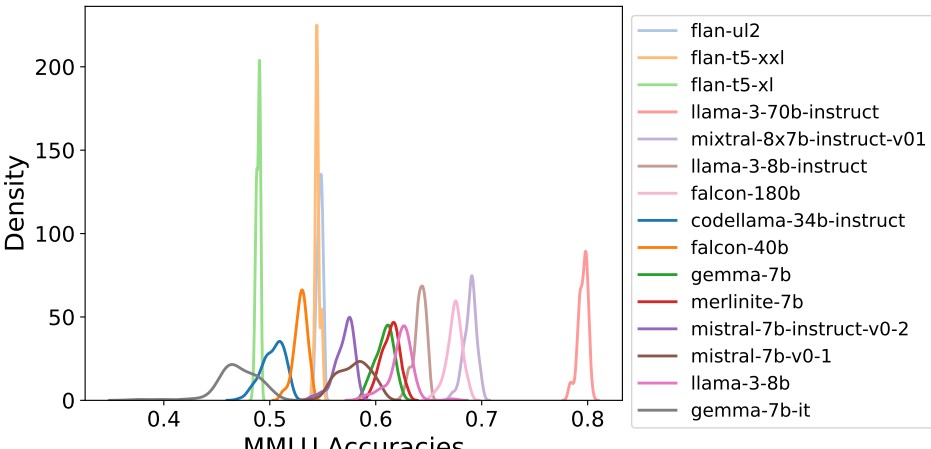

Figure 16: MMLU accuracy (all 57 subjects).

To correlate the ranks from different judges, we can use Kendall's $W$. Kendall's $W$ [Kendall and Smith, 1939] ranges from 0 (no agreement) to 1 (perfect agreement) and is calculated as $W = \frac{12S}{m^2(n^3-n)}$, where $S$ is the sum of squared deviations of the total ranks from the mean rank, $m$ is the number of rankers, and $n$ is the number of objects ranked. In our case, we first have MMLU subjects ranking prompt templates, and then we have LLMs ranking prompt templates.

In Figure 17, we see the distribution of Kendall's $W$ for subjects ranking templates. The correlation is not significant, with the highest $W$ around 0.25. This suggests that there is no "best" prompt for a subject.

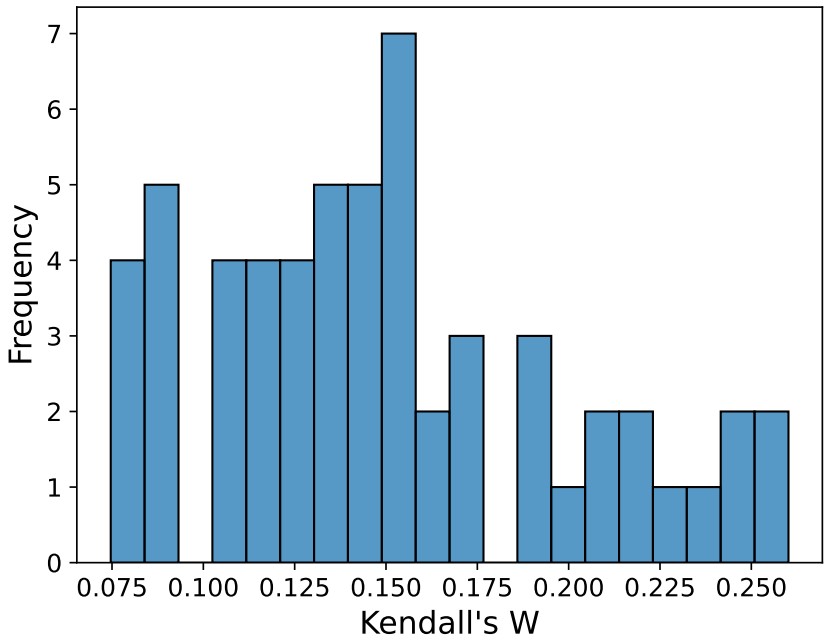

Figure 17: Kendall's $W$ per MMLU subject: Here we see a distribution of Kendall's $W$ over the 57 subjects of MMLU.

In Table 1, we see the values of Kendall's $W$ for each model. For most models, the $W$ value is not high, but for gemma-7b and mistral-7b-v0-1, the value of $W$ is 0.45 and 0.35, respectively. Curiously, both of the top-ranked prompt templates have lots of commas. The best-ranked prompt is **"The, following, are, multiple, choice, questions, (with, answers), about, topic], question], Answers], choices], Answer]"**. Interestingly, the comma separation of each word or phrase in this prompt template may aid the model in parsing and effectively understanding the different components of the prompt structure.

Table 1: Kendall's $W$ per LLM

| Model | Kendall's W |
|---|---|
| meta-llama/llama-3-8b-instruct | 0.126027 |
| meta-llama/llama-3-8b | 0.252835 |
| meta-llama/llama-3-70b-instruct | 0.101895 |
| mistralai/mistral-7b-instruct-v0-2 | 0.219841 |
| mistralai/mistral-7b-v0-1 | 0.345592 |
| mistralai/mixtral-8x7b-instruct-v01 | 0.131487 |
| codellama/codellama-34b-instruct | 0.287066 |
| ibm-mistralai/merlinite-7b | 0.146411 |
| **google/gemma-7b-it** | **0.445478** |
| google/gemma-7b | 0.179373 |
| google/flan-t5-xl | 0.066501 |
| google/flan-t5-xxl | 0.056257 |
| google/flan-ul2 | 0.109076 |
| tiiuae/falcon-180b | 0.165600 |
| tiiuae/falcon-40b | 0.100173 |

Figure 18 illustrates sensitivity for llama-3-8b, gemma-7b, and merlinite-7b, respectively. On the template graph, a distance 1 means templates differ by only 1 feature, a distance 2 means templates differ by 2 features, etc. We see that there is no significant correlation between template distance and the accuracy spread. In the cases of gemma-7b and merlinite-7b, the accuracy spread for templates with smaller distance seems to be smaller, possibly implying that the template graph for these models is smooth.

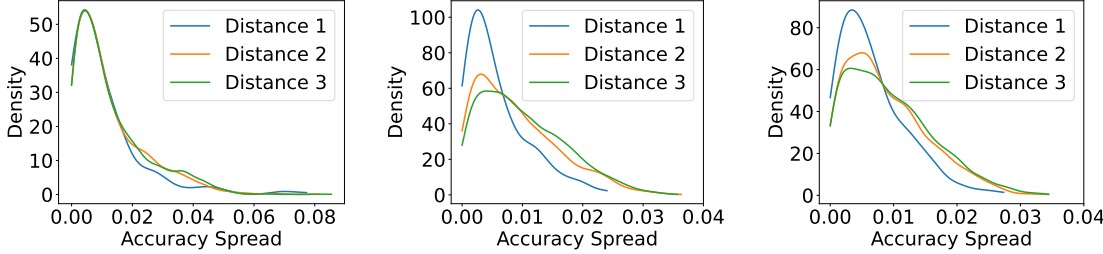

Figure 18: Model sensitivity for llama-3-8b, gemma-7b, and merlinite-7b.

## L BERT fine-tuning details

### L.1 Model

We augment the BERT model by extending its input embeddings by $|J|$ [Example ID] tokens which we use to feed information about the example identity to the model. Additionally, we add a linear

downward projection ($d = 25$) on top of the final BERT layer to reduce the dimensionality of the resulting covariates.

## L.2 Training data

To obtain training examples, we concatenate all prompting templates with all [Example ID] tokens giving us $|I| \times |J|$ model inputs (giving us the following dataset sizes for the respective benchmarks: BBH 209,280; LMentry 175,776; and MMLU 1,121,568). Labels consist of vectors of correctness scores $y_{ij}$ from the LLMs in the training set, making the training task a multi-label binary classification problem. We train on an iid split of half of the LLMs at a time and test on the other half. Additionally, the training data are split along the example axis into an 80% training and 20% validation set.

## L.3 Hyperparameters

We run a small grid search over different plausible hyperparameter settings and settle on the following setup: We employ the Adam optimizer [Kingma and Ba, 2014] with an initial learning rate of 2e-5 and a weight decay of 1e-5. The learning rate undergoes a linear warm-up over 200 steps, followed by exponential decay using the formula $lr_{currnt} = \gamma^s \cdot lr_{init}$, where $s$ is the number of steps after the warmup phase and the decay factor $\gamma$ is set to 0.99995. We train with a batch size of 96.

## M Heuristics for discrete features

For the BBH and LMentry benchmarks, we use the following heuristics to construct feature representations of prompt templates.

Table 2: Overview of Discrete Features

| Category | Feature Name | Description |
| --- | --- | --- |
| Casing Features | All Caps Words | Count of all uppercase words |
| | Lowercase Words | Count of all lowercase words |
| | Capitalized Words | Count of words with the first letter capitalized |
| Formatting Features | Line Breaks | Count of line breaks |
| | Framing Words | Count of capitalized or numeric words before a colon |
| Special Characters Features | Colon (:) | Count of ':' |
| | Dash (-) | Count of '-' |
| | Double Bar (‖) | Count of '‖' |
| | Separator token | Count of '<sep>' |
| | Double Colon (::) | Count of '::' |
| | Parenthesis Left (() | Count of '(' |
| | Parenthesis Right ()) | Count of ')' |
| | Quotation (") | Count of '"' |
| | Question Mark (?) | Count of '?' |
| Length Feature | Space Count | Count of spaces |

