# OpenReview forum: "Efficient multi-prompt evaluation of LLMs"
_NeurIPS.cc/2024/Conference — NeurIPS 2024 poster_

### Official Review · Reviewer_emjz · 2024-07-03

**Soundness:** 3
**Presentation:** 3
**Contribution:** 3
**Rating:** 8
**Confidence:** 1

**Summary:**

The authors introduce a novel method called PromptEval which permits efficient multi-prompt evaluation of LLMs across different prompt templates with a limited number of evaluations.

**Strengths:**

- Theoretical guarantees that PromptEval has desirable statistical properties such as consistency in estimating performance distribution and its quantiles.
- Large-scale study of prompt sensitivity of 15 popular open-source LLMs on MMLU.

**Weaknesses:**

N/A

**Questions:**

N/A

---

> ### Author Rebuttal · Authors · 2024-07-31
>
> Dear reviewer, thank you for your comments.

---

### Official Review · Reviewer_hMGK · 2024-07-12

**Soundness:** 3
**Presentation:** 2
**Contribution:** 3
**Rating:** 5
**Confidence:** 3

**Summary:**

This paper introduces PromptEval, a novel method for efficiently evaluating the performance of large language models (LLMs) across multiple prompt templates. They propose a statistical framework based on Item Response Theory (IRT) for estimating LLM performance distribution across many prompts using limited evaluations and theoretical guarantees on the consistency of the proposed estimators.

**Strengths:**

1. The paper addresses an important and timely problem in LLM evaluation, recognizing the limitations of single-prompt evaluations and proposing a novel solution.
2. The theoretical foundation is solid, with clear assumptions and complete proofs provided in the appendix.

**Weaknesses:**

1. While the authors compare different variations of their method, they don't explore alternative modeling approaches beyond IRT. It would be interesting to see comparisons with other potential frameworks.
2. While the method saves evaluations, the paper doesn't discuss the computational requirements of the estimation process itself, which could be relevant for very large prompt sets.

**Questions:**

1. How sensitive is PromptEval to the choice of initial prompt set? How might this impact its use in practice?
2. How does the computational complexity of the estimation process scale with the number of prompts and examples?

**Limitations:**

The authors have provided a comprehensive discussion of limitations in the appendix, which is commendable.

---

> ### Author Rebuttal · Authors · 2024-07-31
>
> Dear reviewer, thank you for your dedication to our paper. We addressed the issues you raised below. Please let us know if you have any other questions.
>
> - **Use of IRT and alternatives:** We use (a generalization of) IRT because it is the model most suited to our data. Although there are other psychometric models (e.g. classical test theory based on Gaussian factor analysis), they are not as suitable as IRT. Is there a specific alternative model that you wish us to comment on?
>
> - **Computational cost of estimation for the experiments in the paper:** The estimation process is pretty cheap. For example, in our experiments, which reflect very well real use cases, we fitted logistic regression models in datasets with less than 2k samples and a couple of hundred (or a few thousand) columns, which is performed in a few seconds by a laptop. We will make this observation clear in the paper.
>
> - **More details on the computational complexity with growing dataset dimensions:** Consider the case of our experiment in which prompts are represented by embeddings of fixed size and examples are represented by one-hot encodings. Because the dimension of the embeddings does not depend on the number of prompt variations, the number of samples and variables used to fit our model does not vary with the number of prompt variations. Then, *computational costs are constant with respect to the number of prompt templates*. On the other hand, the number of variables (and consequently samples, to make estimation possible) should increase linearly with the number of examples, which are usually hundreds or a few thousand. Thus, this should not be a problem in most practical cases. We will add a comment about this point in the paper. Thank you for bringing this up!
>
> - **Sensitivity of PromptEval:** Our method aims to estimate performance distribution across a **given prompt set**, which can be obtained using methods from [1,2], for example. We anticipate the method to work well with any initial prompt set in the sense of accurately estimating performance distribution across the given prompt set, as supported by experiments considering different ways to generate the initial prompts. We note that if the initial prompt set is too limited or biased in some sense, the resulting estimate of performance distribution might not accurately reflect sensitivity to **all possible prompts**. We make this point clearer in our "Limitations" section. Thank you!
>
>
> **References**
>
> [1] Moran Mizrahi, Guy Kaplan, Dan Malkin, Rotem Dror, Dafna Shahaf, and Gabriel Stanovsky. State of what art? a call for multi-prompt LLM evaluation. arXiv preprint arXiv:2401.00595, 2023.
>
> [2] Melanie Sclar, Yejin Choi, Yulia Tsvetkov, and Alane Suhr. Quantifying language models’ sensitivity to spurious features in prompt design or: How i learned to start worrying about prompt formatting. arXiv preprint arXiv:2310.11324, 2023.

---

> > ### Comment · Reviewer_hMGK · 2024-08-12
> > **Reply by Reviewer**
> >
> > Thank you for your detailed rebuttal. Your response has addressed my major concerns. I will also engage in further discussion with the other reviewers to ensure we consider your clarifications.

---

### Official Review · Reviewer_PhKA · 2024-07-13

**Soundness:** 3
**Presentation:** 3
**Contribution:** 3
**Rating:** 6
**Confidence:** 4

**Summary:**

This paper introduces PromptEval, an efficient multi-prompt evaluation method for LLMs, showing its statistical consistency and effectiveness across benchmarks (MMLU, BBH, LMentry) and studying prompt sensitivity in 15 open-source LLMs.

**Strengths:**

- The authors conducted a comprehensive theoretical analysis to verify that PromptEval possesses ideal statistical properties, and carried out extensive experiments using three benchmarks (MMLU, BBH, LMentry) and fifteen open-source large language models (flan series, llama series, mistral series, etc.)
- They introduces five variations, comparing the baseline under different budgets (i.e., different sampling rates) and various quantiles, demonstrating the significant effectiveness of PromptEval.

**Weaknesses:**

- The current experiments have only been conducted on open-source models. I look forward to experiments on some mainstream closed-source models (such as the GPT series or the Claude series) and explore corresponding variant strategies.
- This method PromptEval requires that a large number of active prompt temples exist in its benchmarks, but this is often not satisfied. Does this method work if there are only a few temples in benchmarks? This requires authors to carry out some experiments to prove.
- The construction of the baseline is relatively simple, and authors need to compare it with some relevant methods (like TRIPLE[1]) to highlight the advantages of PromptEval.

[1] Chengshuai Shi, Kun Yang, Jing Yang, and Cong Shen.  Efficient Prompt Optimization Through the Lens of Best Arm Identification, arXiv preprint arXiv: http://arxiv.org/abs/2402.09723.

**Questions:**

Please refer to my comments in "Weaknesses".

**Limitations:**

The author pointed out the limitations in the Appendix.

---

> ### Author Rebuttal · Authors · 2024-07-31
>
> Dear reviewer, thank you for your work on our paper. We addressed the issues you raised below. Please let us know if you have any questions.
>
> - **New experiment with closed-source models:** We have conducted a new experiment using closed-source models. Due to the high costs associated with running these models (such as GPT-4), we carried out a small-scale experiment. Moreover, to enhance the diversity of experiments and interest of our paper, we explored the concept of LLM-as-a-judge. In this setup, we generated various prompt templates to present to the judge, which in this case is a closed-source model (GPT-4o-mini). This approach allows us to assess how sensitive evaluated models' performance is to different evaluation prompts. Specifically, we used AlpacaEval 2.0 [1] as the benchmark, generated 100 prompt variations using ChatGPT, and presented these to the judge. We evaluated the performance of four LLMs with similar capabilities (cohere, Qwen1.5-7B-Chat, Mistral-7B-Instruct-v0.2, llama-2-70b-chat-hf) using only 2% of the total evaluations (1.6k/80k). The new results have been included in the additional PDF submitted through OpenReview. In summary, we show that PromptEval offers much superior performance in terms of estimation error (Wasserstein distance $W_1$) when compared with the baseline. Please feel free to reach out if you have any questions.
>
> - **New experiment with fewer prompt variations:** We repeat the main experiment in the paper (randomly) cutting the number of prompt templates by a factor of 5. This means we use only 20 prompt variations in MMLU, for example. In summary, PromptEval still does pretty well, beating the baseline. However, the gap between PromptEval and the baseline has shortened due to fewer variations. This fact highlights that the bigger the number of templates, the more useful PromptEval can be relative to the baseline. We include the new version of Figure 2 in the extra pdf submitted through OpenReview, which summarizes well the new results.
>
> - **Baselines:** We compared against TRIPLE-GSE (the best version of TRIPLE, designed to run when many prompt templates are available) in our “best prompt identification” experiment. Please check the end of Section 5, where we show that PromptEval outperforms TRIPLE in **best prompt identification** in MMLU. We do not compare against TRIPLE in the **performance distribution** estimation experiment because that method is not designed for that purpose. Currently, as far as we know, PromptEval is the only method designed for efficient estimation of performance distribution across prompt templates (besides the naive averaging 'avg', included in our experiments).
>
> **References**
>
> [1] Li, Xuechen, et al. "Alpacaeval: An automatic evaluator of instruction-following models." (2023).

---

### Author Rebuttal · Authors · 2024-08-03

Dear reviewers,

Thank you for your time reviewing our paper. We include new experiments suggested by reviewer PhKA in the extra pdf.
1. In the first experiment, we explore the concept of LLM-as-a judge. We used PromptEval to estimate the distribution of performances given by a closed-source LLM (GPT-4o-mini, the "judge"), across 100 evaluation prompt templates, when evaluating other LLMs responses on AlpacaEval 2.0. Realize that, contrasting with previous experiments, we do not rephrase prompts given to the models being evaluated but only to the evaluator. This approach allows us to assess how sensitive evaluated models' performance is to different evaluation prompts. As a result, we show that PromptEval is also valuable for more robust LLM-as-a-judge benchmarks.
2. In the second experiment, we wanted to check how PromptEval performs when fewer prompt variations are available. Then, we repeat the main experiment of the paper only keeping 20% (randomly chosen) of the total number of prompts. In summary, PromptEval still does pretty well, beating the baseline. However, the gap between PromptEval and the baseline has shortened due to fewer variations. This fact highlights that the bigger the number of templates, the more useful PromptEval can be relative to the baseline.

---

### Decision · Program_Chairs · 2024-09-25

**Decision:**

Accept (poster)

**Comment:**

The paper argues that a robust evaluation of LLM performance should be conducted on a distribution of prompts and discusses a sample-efficient way to obtain such an evaluation. For the proposed problem, the paper makes a solid contribution by (1) proposing an efficient estimation of such a measurement with theoretical guarantees under certain assumptions, and (2) conducting comprehensive experiments on open-source LLMs, which leads to some interesting findings. Reviewers also like the proposed direction and only had some minor comments.

The AC, however, is worried that the proposed distribution-based measurement may not be the best measurement to use because:

(1) The AC thinks it's better to measure both "robustness" (how an LLM performs under various different prompts for the same task) and "performance" (the performance of an LLM with the best prompt). There are many cases where only the latter (peak performance) is important since we want to deploy a model with good performance. The former may be used in Q/A-typed queries by general users, where users may not try to find the best prompt. The paper will provide a good measurement for the former but not the latter.

(2) As also pointed out by a reviewer, it is unclear whether the provided set of templates is complete. So even though the paper measures performance under a distribution of templates, it can still be biased due to the template family used in the evaluation.

Given that the paper is making a solid contribution to measuring the "robustness" perspective, the AC would still recommend acceptance but suggests the authors revise the paper to state this limitation of the work clearly in the introduction.